# Metabolism of multiple glycosaminoglycans by *Bacteroides thetaiotaomicron* is orchestrated by a versatile core genetic locus

Didier Ndeh[1,2], Arnaud Baslé[1], Henrik Strahl[3], Edwin A. Yates[4], Urszula L. McClurgg [4], Bernard Henrissat[5,6,7], Nicolas Terrapon [5,6] & Alan Cartmell [4]*

The human gut microbiota (HGM), which is critical to human health, utilises complex glycans as its major carbon source. Glycosaminoglycans represent an important, high priority, nutrient source for the HGM. Pathways for the metabolism of various glycosaminoglycan substrates remain ill-defined. Here we perform a biochemical, genetic and structural dissection of the genetic loci that orchestrates glycosaminoglycan metabolism in the organism *Bacteroides thetaiotaomicron*. Here, we report: the discovery of two previously unknown surface glycan binding proteins which facilitate glycosaminoglycan import into the periplasm; distinct kinetic and genetic specificities of various periplasmic lyases which dictate glycosaminoglycan metabolic pathways; understanding of endo sulfatase activity questioning the paradigm of how the 'sulfation problem' is handled by the HGM; and 3D crystal structures of the polysaccharide utilisation loci encoded sulfatases. Together with comparative genomic studies, our study fills major gaps in our knowledge of glycosaminoglycan metabolism by the HGM.

[1] Biosciences Institute, Newcastle University, Newcastle upon Tyne NE2 4HH, UK. [2] Quadram Institute Bioscience, Norwich Research Park, Norwich, Norfolk NR4 7UQ, UK. [3] Centre for Bacterial Cell Biology, Biosciences Institute, Newcastle University, Newcastle upon Tyne NE2 4HH, UK. [4] Department of Biochemistry, Institute of Integrative Biology, University of Liverpool, Liverpool L69 7ZB, UK. [5] Architecture et Fonction des Macromolécules Biologiques, CNRS, Aix-Marseille University, F-13288 Marseille, France. [6] USC1408 Architecture et Fonction des Macromolécules Biologiques, Institut National de la Recherche Agronomique, F-13288 Marseille, France. [7] Department of Biological Sciences, King Abdulaziz University, Jeddah 23218, Saudi Arabia.
*email: Alan.Cartmell@liverpool.ac.uk

                                                                        1

The human gut microbiota (HGM) is a vast microbial community that is crucial to human health[1]. The maintenance of the HGM is dependent on its ability to utilise complex glycans as a nutrient source, in the form of dietary and host glycans[2]. The mechanisms by which this community metabolises these glycans is therefore critical to understanding the ecology of the system and how it could be manipulated to benefit human health. Bacteroidetes is one of the two major phyla that dominate the HGM[3]. These organisms dedicate up to 20% of their genome to complex glycan metabolism in the form of genes primarily encoding carbohydrate active enzymes such as, polysaccharide lyases (PLs), glycoside hydrolases, glycan transport and binding proteins, and carbohydrate sulfatases. Bacteroides arrange these genes into polysaccharide utilisation loci (PUL) which are co-localised, co-regulated genes, in response to a specific glycan[4,5]. The vast majority of our understanding of how Bacteroidetes metabolise complex glycans is based on dietary plant glycans[6–9] yet host glycans represent a substantial carbon reservoir, which is also accessed by the HGM[10,11]. Glycosaminoglycans (GAGs) have been demonstrated to be a high-priority carbon source for the HGM organism Bacteroides thetaiotaomicron (B. theta)[11,12], and although a pathway for Heparin/Heparan sulfate[13] (H/HS) has been described, the metabolism of the complex, and distinct, GAGs chondroitin sulfate (CS-A and CS-C), dermatan sulfate (DS) and hyaluronic acid (HA) is poorly described. Enzymes of the HGM that metabolise CS have been described[13–15], yet how they co-ordinate together, their mechanistic details, mode of action and sulfation tolerances remain largely opaque. In addition, the role and mechanism of action of the surface glycan binding proteins (SGBPs) in the degradation of CS, DS and HA is unknown. Here we report how a core PUL has evolved to metabolise the complexity and heterogeneity of CS-A, CS-C, DS and HA. The $PUL_{CS/DS/HA}$ is co-regulated with distant genes, including a hyaluronidase belonging to the recently classified PL33[16], to complement its core activities and thus fully enabling multiple GAG breakdown. Crystal structures of the two PUL-encoded carbohydrate sulfatases provide key insights into substrate recognition by these critical, and therapeutically relevant[17], enzymes. The data enable a general model to be proposed for the metabolism of CS, DS and HA by the HGM and the Bacteroidetes phylum in a wider context.

## Results

**Structure of CS-A, CS-C, DS and HA**. CS-A and CS-C are both composed of repeating disaccharide units composed of D-glucuronic acid (GlcA) and N-acetyl-D-galactosamine (GalNAc). GlcA is linked β(1,3) to GalNAc which is then linked β(1,4) to the next GlcA. CS-A and CS-C differ only in sulfation with CS-A highly enriched in O4 sulfation on GalNAc whilst CS-C is enriched in O6 sulfation on GalNAc. DS is related to CS-A except GlcA is predominantly replaced with L-iduronic acid which is connected to GalNAc via an α(1,4) linkage. Finally, HA is similar to CS-A but replaces GalNAc with N-acetyl-D-glucosamine (GlcNAc) and contains no sulfation (Fig. 1b).

**$PUL_{CS/DS/HA}$ structure**. B. theta grows on CS-A, CS-C, DS and HA and transcriptomics, on CS-C and HA, demonstrate only $PUL_{CS/DS/HA}$ and two additional genes, $BT1596-S1\_9^{2S-sulf}$ and $BT4410^{PL33}$ are upregulated[4,13]. $PUL_{CS/DS/HA}$ encodes three PLs ($BT3324^{PL8}$, $BT3328^{PL29}$ and $BT3350^{PL8}$), one GH88 ($BT3348^{GH88}$), 2-O-sulfatases ($BT3333-S1\_15^{6S-sulf}$ and $BT3349-S1\_27^{4S-sulf}$), a single SusC/D-like transporter system ($BT3331^{susD}$-$BT3332^{susC}$), a hybrid two-component sensor ($BT3334^{HTCS}$) and two proteins of unknown function (BT3329 and BT3330). The non-PUL encoded genes, $BT1596-S1\_9^{2S-sulf}$

and $BT4410^{PL33}$, encode a 2-O sulfatase and a hyaluronidase, respectively (Fig. 1a, b). A paradigm for how PULs organise their protein apparatus has been established. A surface enzyme(s), with usually one SGBP, degrades and captures large polysaccharides/oligosaccharides at the cell surface. These partially degraded polysaccharides/oligosaccharides are transported into the periplasm through the action of the highly conserved, and critical, SusC/D-like transporter system. Small polysaccharides/large oligosaccharides transported into the periplasm are then metabolised to their constituent parts, usually monosaccharides, and transported into the cytoplasm to enter cytoplasmic metabolic pathways.

**Extracellular glycan binding and degradation**. Two genes of unknown function BT3329 and BT3330 exist within $PUL_{CS/DS/HA}$. Both proteins had no enzymatic activity against any GAG substrates. However, they were shown to exhibit binding by isothermal titration calorimetry (ITC). Both proteins were able to bind all four GAGs and were thus classified as SGBPs (Supplementary Fig. 1 and Supplementary Table 1). $BT3329^{SGBP}$ bound to CS-A and HA with approximately ten and fourfold higher affinity than $BT3330^{SGBP}$ whilst against CS-C and DS $BT3329^{SGBP}$ bound with an ~100 and ~50-fold higher affinity. $BT3329^{SGBP}$ but not $BT3330^{SGBP}$ bound a CS oligosaccharide of DP10. $BT3330^{SGBP}$ was, however, able to bind a CS oligosaccharide of DP20. The SGBPs thus differ in their polysaccharide and oligosaccharide preferences. $BT3329^{SGBP}$ has the ability to bind more variably sulfated and smaller glycans than $BT3330^{SGBP}$, and displays a greater tolerance to iduronic acid (Supplementary Fig. 1 and 2a). Both proteins were confirmed to be extracellular by immunofluorescence assays (Supplementary Fig. 2b). $BT3331^{susD}$ demonstrated a very strict substrate specificity binding only to CS-A with a measurable affinity, similar to that of $BT3329^{SGBP}$ (Supplementary Table 1). The surface lyase, $BT3328^{PL29}$, showed its highest enzymatic activity and affinity against CS-A > CS-C > HA and little activity on DS, differing from previous data[18]. End point assays, the total number of glycosidic bonds cleaved, followed the relationship of HA > CS-C > CS-A (Fig. 2 and Supplementary Table 2). This demonstrates that although O4 sulfation increases $BT3328^{PL29}$ activity on CS-A it also limits glycosidic bond access in some contexts. $BT3328^{PL29}$ was only able to cleave HA oligosaccharides with a DP ≥ 8, indicating the enzyme has at least eight subsites (Supplementary Fig. 4a). Interestingly, $BT3328^{PL29}$ displays very similar substrate specificities to $BT3331^{susD}$, which, could suggest that the proteins work closely together at the cell surface. When $BT3328^{PL29}$ was incubated with HA oligosaccharides of d.p. 12, 10 and 8, labelled at their reducing end with a fluorescent tag, a labelled trisaccharide was produced from d.p. 12 substrate and a labelled disaccharide was produced from d.p. 10 and 8 substrates. These data indicate that $BT3328^{PL29}$ cleaves glycans from the reducing end (Supplementary Fig. 2c, d and Supplementary discussion). We identified a conserved region of ~300 amino acids at the N-terminal of $BT3328^{PL29}$ and $BT3329^{SGBP}$. This region was also shared with SGBP-positioned genes from other PULs, which do not share a detectable sequence similarity downstream of this ~300 amino acid at the N-terminal region, like $BT3328^{PL29}$ and $BT3329^{SGBP}$. This might constitute a 'spacer' domain, not homologous but, functionally equivalent to the BACON domain found at the N-terminal region of the surface-anchored GH5 in the xyloglucan PUL[7] and in many other SGBPs. Such domains would allow extracellular proteins to extend out of the capsular envelope and/or increase their mobility.

**Protein–protein interactions**. The interaction of the cell surface proteins with one another was assessed by Native PAGE and

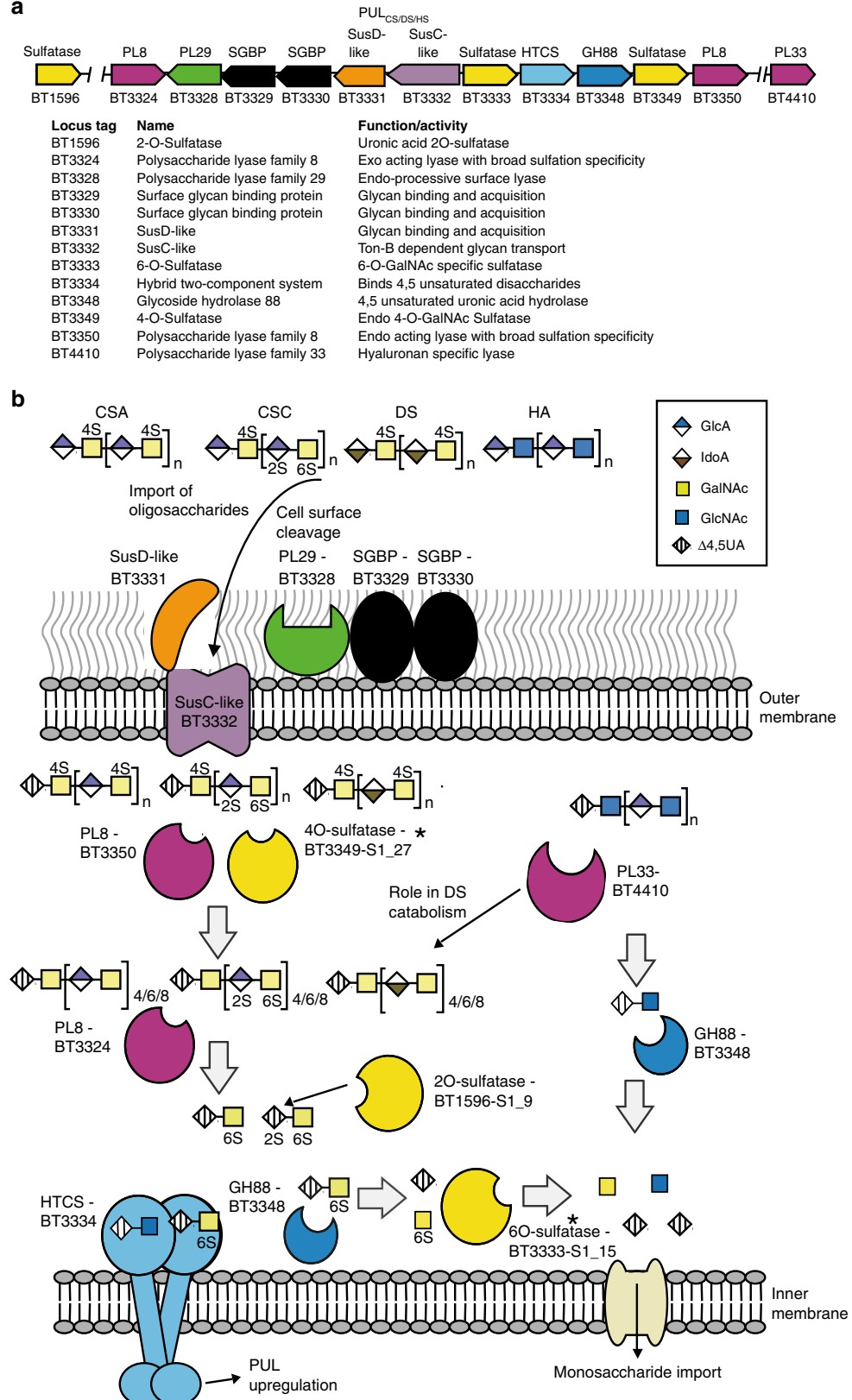

**Fig. 1 Model of glycosaminoglycan degradation by PUL$_{CS/DS/HA}$. a** Organisation of the PUL$_{CS/DS/HA}$ locus in *Bacteroides thetaiotaomicron*. **b** Schematic depiction of the activity, location and specificity of PUL$_{CS/DS/HA}$ encoded enzymes. Asterisk indicates structures were solved in this study.

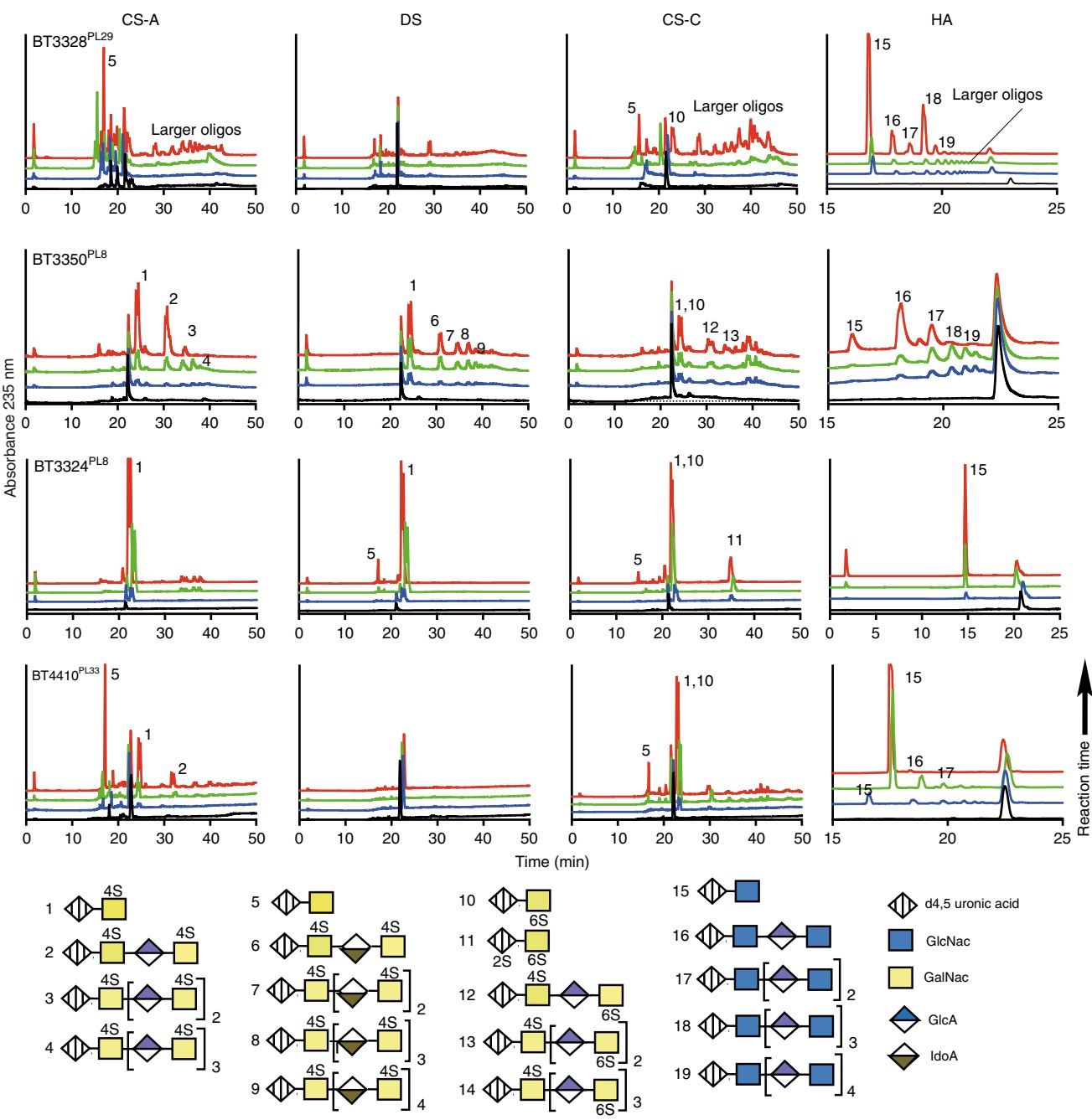

**Fig. 2 HPLC product profiles of the PUL$_{CS/DS/HA}$ encoded lyases.** Peaks were identified by comparison with commercially purchased standards. The black, blue, green and red lines represent 0, early, mid-point and late stages of the reaction time course for each enzyme (CS-A chondroitin sulfate A, DS dermatan sulfate, CS-C chondroitin sulfate C and HA hyaluronic acid). All reactions were carried out in 50 mM MES, pH 6.5, with 150 mM NaCl at 37 °C.

co-immunoprecipitation (Co-IP). Native PAGE demonstrated that BT3328[PL29], BT3329[SGBP], BT3330[SGBP] and BT3331[SusD] displayed no observable interaction in vitro (Supplementary Fig. 3a, b). Co-IP data, however, demonstrated that at the cell surface BT3328[PL29], BT3329[SGBP] and BT3330[SGBP] do indeed interact (Supplementary Fig. 3c–e). Co-IPs using anti-BT3328[PL29] and anit-BT3329[SGBP] antibodies revealed the presence of both BT3329[SGBP] and BT3330[SGBP] in BT3328[PL29]-IP and the presence of both BT3328[PL29] and BT3330[SGBP] in BT3329[SGBP]-IP. When the BT3330[SGBP] precipitant was analysed only BT3329[SGBP] could be observed and not BT3328[PL29].

Collectively these data suggest that BT3328[PL29], BT3329[SGBP] and BT3330[SGBP] interact with each other at the bacterial cell surface in vivo. This interaction requires either membrane attachment or the presence of the integral membrane protein BT3332[SusC].

**Periplasmic glycan degradation apparatus.** *Biochemical analysis of the periplasmic lyases:* BT3324[PL8] and BT3350[PL8] are both PL8s whilst BT4410 has recently been classified into family PL33. BT3350[PL8] is a broad specificity lyase having similar activity on CS-A, CS-C and DS but had reduced activity on HA

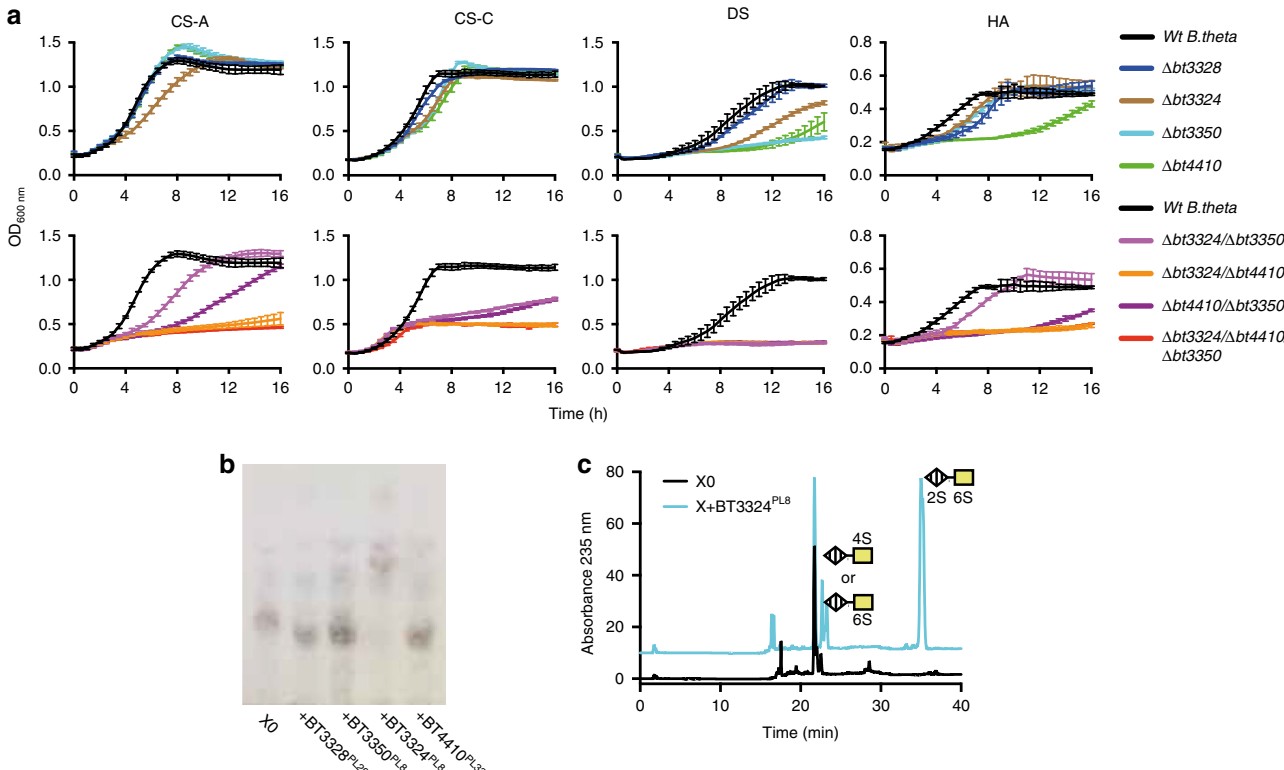

**Fig. 3 Analysis of genetic knockouts and their effect on glycosaminoglycan metabolism. a** Growth of genetic knockouts on glycosaminoglycan substrates (CS-A chondroitin sulfate A, DS dermatan sulfate, CS-C chondroitin sulfate C and HA hyaluronic acid); **b** TLC analysis of purified oligosaccharide (X), left behind in media from Δbt3324 growths, against PUL$_{CS/DS/HA}$ encoded lyases; **c** HPLC analysis of X treated with BT3324$^{PL8}$. Data represent technical triplicates and source data are provided in the source data file labelled as mutant growths.

(Supplementary Table 2). BT3350$^{PL8}$ displayed an endo mode of action but disaccharides were observed earlier in the reaction time course on the sulfated substrates (Fig. 2). End point assays revealed that BT3350$^{PL8}$ could cleave similar amounts of glycosidic linkages in CS-A and DS but around half as much in the GAGs CS-C and HA (Supplementary Table 2). The end products of BT3350$^{PL8}$ HA degradation are mainly of DP2, 4 and 6, with oligos of ≥DP4 making up the majority of the population (Fig. 2). Against CS-A the end products were mainly DP2 and DP4, whilst against CS-C and DS a greater quantity of products >DP6 were observed. Surprisingly, BT3350$^{PL8}$ was able to cleave HA oligos down to disaccharides (Supplementary Fig. 4a). BT3324$^{PL8}$ is an exo-acting lyase releasing exclusively disaccharides from all the GAGs tested (Fig. 2 and Supplementary Fig. 4a). BT3324$^{PL8}$ displayed its highest activity against CS-A and DS with much reduced activities against CS-C and HA. Thus, O4 sulfation is required for maximal activity. BT4410$^{PL33}$ displayed a highly endo-processive mode of action and displayed >100-fold preference for HA. End point assays showed the presence of sulfation limited substrate accessibility, consistent with BT4410$^{PL33}$ being tailored towards HA (Fig. 2 and Supplementary Table 2).

*Genetic analyses of the PUL$_{CS/DS/HA}$ lyases:* The contribution of each lyase in GAG metabolism was further assessed through growth experiments with single, double and triple genetic mutants. Using CS-A and CS-C as the substrate, we observed that individual PL mutants exhibited very little evidence of growth defects; Δbt3324 exhibited a slight defect in growth rate on CS-A (Fig. 3a). The Δbt3350/Δbt4410 showed a stronger phenotype than Δbt3324/Δbt3350 on CS-A, whilst on CS-C both mutants exhibited the same level of defect. This is due to O6 and/or O2 sulfation in CS-C being inhibitory to BT4410$^{PL33}$. Whilst for CS-A the endo 4S sulfatase, BT3349-S1_27$^{4S-sulf}$ (see

'O-sulfatases' section), is able to remove 4S sulfation thus allowing enhanced activity of BT4410$^{PL33}$ on CS-A (Further detail in Supplementary discussion). The double-mutant Δbt3324/Δbt4410 showed extremely limited or no growth on all substrates, comparable with the triple-mutant Δbt3324/Δbt3350/Δbt4410 (Fig. 3a). The Δbt3324/Δbt4410 mutant was previously shown to have drastically reduced BT3333-S1_15$^{6S-sulf}$ mRNA levels in the presence of CS[13], thus, BT3350$^{PL8}$ alone cannot generate disaccharides at a significant rate as to upregulate the PUL. BT3350$^{PL8}$ must, therefore, work in conjunction with BT3324$^{PL8}$ and BT4410$^{PL33}$ to complete GAG catabolism to their disaccharide repeating units. These data are consistent with BT3350$^{PL8}$ generating new non-reducing ends for BT3324$^{PL8}$ and BT4410$^{PL33}$ to act on, demonstrating an endo–exo synergy as commonly observed in other glycan-degrading systems. When HA is the substrate, all mutants that show a defect involved the deletion of BT4410$^{PL33}$. The Δbt4410 mutant by itself was sufficient to cause a severe growth defect, whilst the Δbt3324/Δbt3350 mutant showed only a minor defect, reinforcing the critical role of BT4410$^{PL33}$ as a hyaluronan specific lyase (Fig. 3a). When DS is the growth substrate, loss of any periplasmic lyase alone is sufficient to cause a significant defect, but loss of either BT4410$^{PL33}$ or BT3350$^{PL8}$ is particularly severe (Fig. 3a). Both enzymes have endo character indicating that the ability to cleave linkages internally in DS is critical to its breakdown by *B. theta*. All double mutants failed to grow at all on DS (Fig. 3a).

*Analyses of spent supernatants from genetic mutants:* Analysis of spent supernatants from mutant growths revealed that the Δbt3324 mutant, produced a unique product. This substrate was purified and treated with the different lyases; only BT3324$^{PL8}$ showed activity (Fig. 3b). The products were analysed by HPLC and two peaks were identified as UA2S-GalNAc6S disaccharide

and either UA-GalNAc4S or UA-GalNAc6S in a ratio of 3:1 (Fig. 3c). The accumulation of this substrate reveals that BT3324$^{PL8}$ is the only enzyme in *B. theta* capable of cleaving regions within CS-C enriched in 2S sulfation.

**O-Sulfatases**. PUL$_{CS/DS/HA}$ encodes two O sulfatases belonging to the sulfatase subfamilies S1_27 and S1_15, the endo 4S sulfatase BT3349-S1_27$^{4S-sulf}$ and the exo-acting 6S sulfatase BT3333-S1_15$^{6S-sulf}$, as classified by the sulfatlas database[19]. BT1596-S1_9$^{2S-sulf}$ is located outside the PUL and belongs to subfamily S1_9. BT3349-S1_27$^{4S-sulf}$ and BT1596-S1_9$^{2S-sulf}$ are active on all O4 and O2 sulfated CS disaccharides, respectively. This is consistent with previous work on these subfamilies[20]. Whilst BT3333-S1_15$^{6S-sulf}$ was only active on GalNAc6S and is the first member to be functionally and structurally characterised within the S1_15 subfamily (Supplementary Fig. 4b, c). BT3349-S1_27$^{4S-sulf}$ contains an SpII signal peptide (Supplementary Table 3) and, due to its endo mode of action, it was hypothesised that the enzyme may be attached to the extracellular surface. Measures of the enzymatic activity at the cell surface by whole-cell assays (see 'Methods') did not result in any degradation of Δ4,5UA-GalNAc4S, a substrate for BT3349-S1_27$^{4S-sulf}$, even after an incubation time of 16 h. However, when the cells were sonicated and the disaccharide incubated with the CFE, Δ4,5UA-GalNAc4S was completely degraded within 1 h (Supplementary Fig. 4d). This confirms BT3349-S1_27$^{4S-sulf}$ is a periplasmic enzyme. BT3333-S1_15$^{6S-sulf}$ also contains an SpII signal peptide but whole-cell assays, using GalNAc6S, indicate that BT3333-S1_15$^{6S-sulf}$ is periplasmic (Supplementary Fig. 4d). This is consistent with the requirement for the periplasmic BT3348$^{GH88}$ to produce its GalNAc6S substrate (Supplementary Fig. 4d).

*Crystal structures of BT3349-S1_27$^{4S-sulf}$ and BT3333-S1_15$^{6S-sulf}$*: Both enzymes are α/β proteins, composed of two domains, a larger N-terminal domain and a smaller C-terminal 'sub-domain' as is typical of formylglycine sulfatases[21]. An inactive form of BT3349-S1_27$^{4S-sulf}$ was crystallised with a CS-A tetrasaccharide, but only a trisaccharide of Δ4,5UA-GalNAc4S-GlcA could be modelled (Fig. 4a, b, Supplementary Fig. 5a). Subsite nomenclature for carbohydrate sulfatases is such that an S site denotes the position of the labile sulfate, which is attached to a 0 subsite sugar. Additional subsites then increase in number toward the reducing end of the glycan, i.e. +1, +2, etc., and decrease in number toward the non-reducing end of the glycan, i.e. −1, −2, etc[22]. Electron density for the −1, 0 and S subsites was well ordered whilst the +1 GlcA was partially disordered and makes no interactions with the protein. A detailed description of the mutants is provided in the supplementary discussion (Supplementary Table 5 and Supplementary Fig. 5e). The endo activity of BT3349-S1_27$^{4S-sulf}$ is conferred by an 'opening up' of either side of the 0 subsite, allowing glycan extension in both the +1 and −1 direction. Structural comparison of HGM, marine and human carbohydrate sulfatases reveals that the linker between β-strand 4 and α-helix 5 is extended, and/or alterations of the C-terminal subdomain (Fig. 4c–h) commonly occur in the exo sulfatases, precluding endo activity. In BT3333-S1_15$^{6S-sulf}$, however, occlusion of potential positive subsites is achieved by an insertion between β-strand 3 and α-helix 4. These features would also prevent extension of the sugar chain beyond the 0 subsite in the endo sulfatases.

The structure of BT3333-S1_15$^{6S-sulf}$ was solved in complex with GalNAc6S and displays a pocket topology (Fig. 5a, Supplementary Fig. 5b). The 0 subsite is made up primarily of polar interactions (Fig. 5b). The axial O4 of the GalNAc (the unique difference of Gal versus Glc-configured substrates, the latter having an equatorial O4) coordinates with O$_δ$2 of D173 and

NH1 of R174 (for further interactions see Supplementary discussion). These interactions probably drive the specificity for GalNAc versus GlcNAc substrates. R174 also coordinates with O1 and the endocyclic ring oxygen through NH$_2$ whilst O3 interacts with the Nε2 of H225A (Fig. 5b). Comparing overlays of BT3333-S1_15$^{6S-sulf}$ with BT4656-S1_11$^{6S-sulf}$, the GlcNAc6S sulfatase encoded by PUL$_{Hep}$, reveals that the two substrates bind perpendicular to one another and thus no interactions are spatially conserved (Fig. 5c). Despite this both enzymes use an Asp and Arg to bind O4 and a His to co-ordinate O3. In BT3333-S1_15$^{6S-sulf}$ these residues are from the N-terminal side of the protein (D173, R174 and H225) whilst in BT4656-S1_11$^{6S-sulf}$ they are from the C-terminal side of the protein (D361, R363 and H447). In addition, when compared with GALNS (4FDJ), the human 6S GalNAc sulfatase, there is no conservation of the sugar-binding residues between the two proteins (Supplementary Fig. 5c–e). Currently no substrate complex of GALNS exists and the specificity determinants remain undescribed.

**GH88 enzyme**. The BT3348$^{GH88}$ is an essential enzyme for CS metabolism[13]. BT3348$^{GH88}$ showed a similar catalytic efficiency against Δ4,5UA-GalNAc, Δ4,5UA-GalNAc6S and Δ4,5UA-GlcNAc but was inactive against Δ4,5UA2S-GalNAc, Δ4,5UA-GalNAc4S, Δ4,5UA-GalNAc4S6S and Δ4,5UA2S-GalNAc4S6S (all β1,3 linked). BT3348$^{GH88}$ was also inactive on the β1,4 linked heparin disaccharide Δ4,5UA-GlcNAc (Supplementary Table 5). BT3348$^{GH88}$ can tolerate O6 sulfation but not O2 or O4 sulfation, and is specific for the β1,3 linkage but has no preference for GlcNAc or GalNAc at its +1 subsite.

**Growth and bioinformatic analyses of gut *Bacteroides* species**. Eleven *Bacteroides* species were grown on each GAG as sole carbon source and classified as either strong, intermediate or non-utilisers. For CS-A and CS-C *B. theta*, *B. ovatus*, *B. eggerthii*, *B. xylanisolvens* and *B. plebeius* were all strong utilisers whilst *B. caccae*, *B. cellulosilyticus*, and *B. intestinalis* demonstrated intermediate growth. When DS was the substrate there was a shift in classifications compared with CS-A and CS-C. *B. eggerthii* and *B. plebeius* switched to being non-utilisers whilst *B. intestinalis* was classified as a strong utiliser. When HA was the substrate *B. ovatus* and *B. eggerthii* out performed all other strains. Only *B. theta* grew as fast but was classified as intermediate due to a ~20% lower final OD$_{600\,nm}$ (Fig. 6).

Orthologous proteins were searched to identify PUL$_{CS/DS/HA}$ relatives in the eleven species. The three strains *B. vulgatus*, *B. dorei* and *B. fragilis* lack PUL$_{CS/DS/HA}$ and only harbour a BT3348$^{GH88}$ orthologue (and a separated BT4410$^{PL33}$ in *B. fragilis*). Consistently, all three showed no growth on any GAGs. We observed BT3328$^{PL29}$-BT3330$^{SGBP}$ were more rapidly diverging and completely lost in *B. cellulosilyticus*, *B. intestinalis* and *B. eggerthii*. However, no impact on growth was observed, consistent with Δ*bt3328* mutant growth data. The only genetic differences with potential growth impacts were the loss of BT4410$^{PL33}$ orthologues in both *B. eggerthii* and *B. plebeius* (which additionally lacks a BT3350$^{PL8}$ orthologue). The importance of BT4410$^{PL33}$ for *B. theta* growth on DS and HA (i) explains the switch of *B. eggerthii* and *B. plebeius* from strong utilisers on CS-A and CS-C to non-utilisers on DS; (ii) suggests that alternative pathways for HA catabolism exist in these organisms compared with *B. theta*. *B. eggerthii* notably outcompeted most other strains on CS-A, CS-C and HA. This could mean members of the PL35 (BACPLE_02279, BACEGG_01448) or PL37 (BACEGG_00231) which are active on GAGs[16,23] and not present in *B. theta*, are contributing to this enhanced growth.

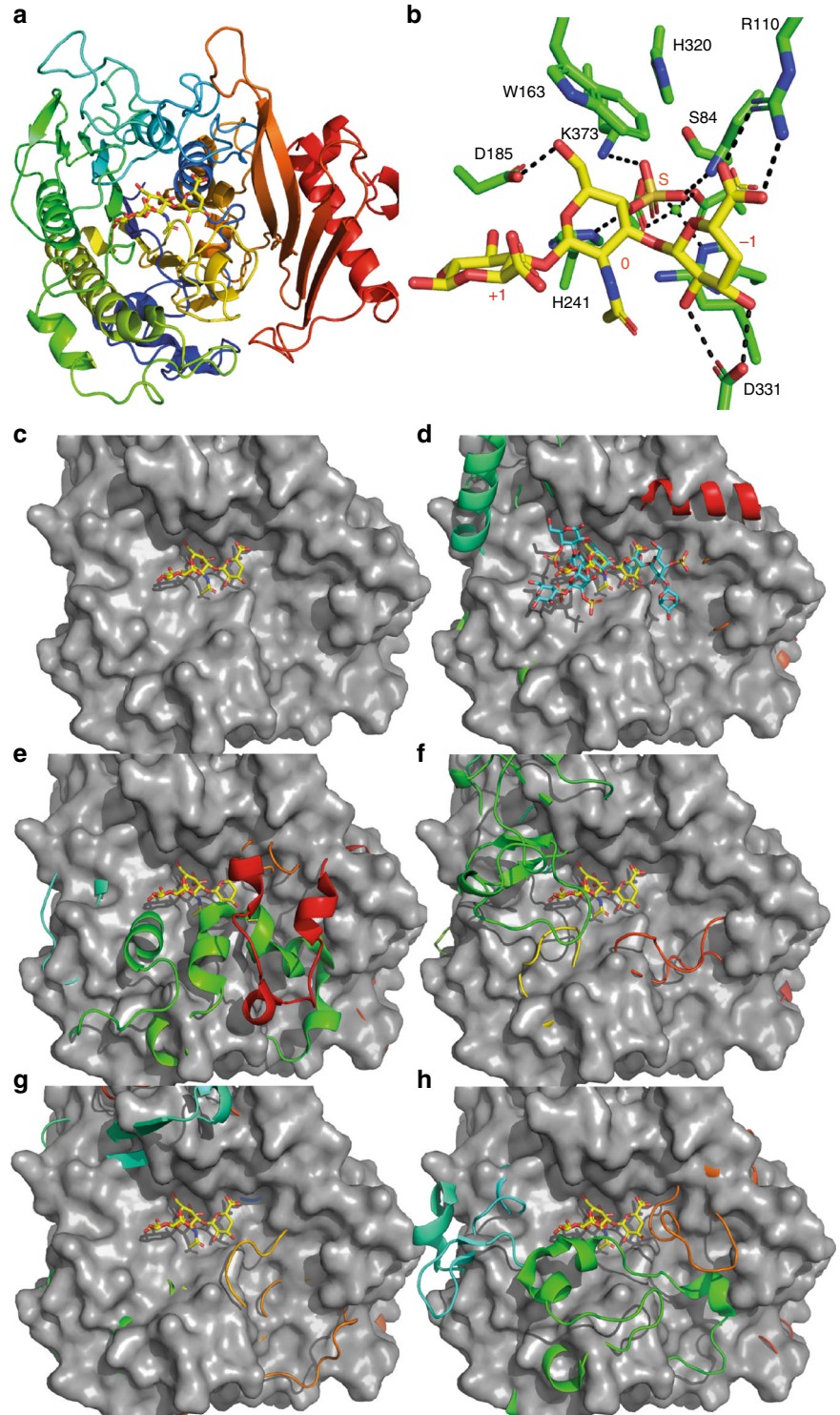

**Fig. 4 Structural depictions of BT3349-S1_27$^{4S}$ and its comparison with other endo and exo sulfatases. a** Cartoon representation of BT3349-S1_27$^{4S\text{-sulf}}$ ramped blue to red from its N- to C-terminus; **b** stick representation of binding site interactions with the protein in green and the bound trisaccharide yellow; **c** surface representation of BT3349-S1_27$^{4S\text{-sulf}}$ showing how the +1 and −1 ends of the trisaccharide are free to be extended; **d** overlays of surface representations of BT3349-S1_27$^{4S\text{-sulf}}$ and a cartoon of the marine endo 4S ɩ-carrageenan sulfatase (PsS1_19A). CS and ɩ-carrageenan oligosaccharides are shown in yellow and cyan, respectively. **e** Overlay of a surface of BT3349-S1_27$^{4S\text{-sulf}}$ with a cartoon of the exo-sulfatase BT1596-S1_9$^{2S\text{-sulf}}$. **f** Overlay of a surface of BT3349-S1_27$^{4S\text{-sulf}}$ with a cartoon of the exo-sulfatase BT3333-S1_15$^{6S\text{-sulf}}$. **g** Overlay of a surface of BT3349-S1_27$^{4S\text{-sulf}}$ with a cartoon of the human exo-sulfatase 1FSU$^{4S\text{-sulf}}$. **h** Overlay of a surface of BT3349-S1_27$^{4S\text{-sulf}}$ with a cartoon of the human exo-sulfatase 5FQL$^{2S\text{-sulf}}$. All cartoon representations are ramped blue to red from their N- to C-terminus.

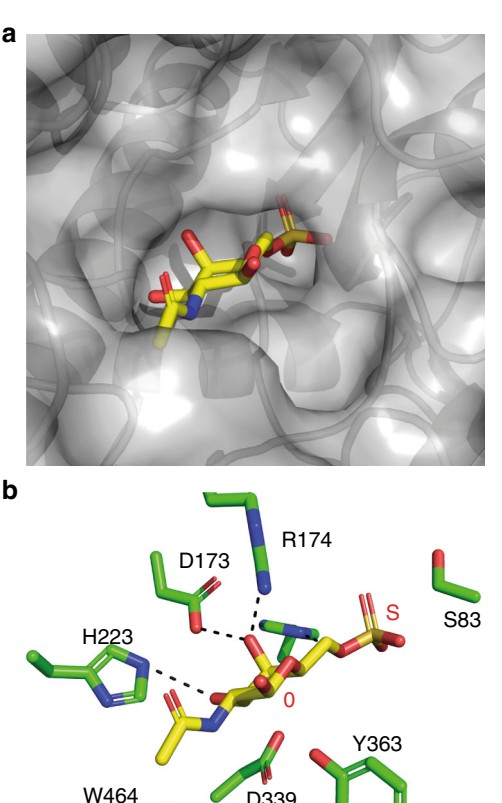

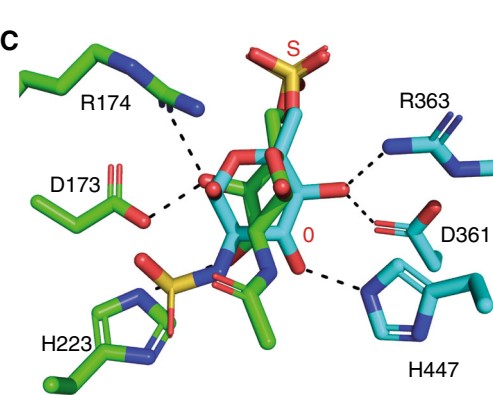

**Fig. 5 Structural depictions of BT3333-S1_15[6S-sulf] and it comparison with BT4656[6S-sulf]. a** A close up of a surface representation showing the pocket topography of BT3333-S1_15[6S-sulf]; **b** stick representations of the binding site interactions with the protein in green and GalNAc6S in yellow; **c** overlay of BT3333-S1_15[6S-sulf] in complex with GalNAc6S (green) and BT4656-s1-11[6S-sulf] in complex GlcNAc6S (cyan).

Comparative genomic analyses on a thousand Bacteroidetes species revealed that PUL$_{CS/DS/HA}$ is largely associated with the *Bacteroides* genus but not ubiquitous. Different sections are variably conserved and may have accessory interest, like the aforementioned BT3328[PL29]-BT3330[SGBP] region (Supplementary Fig. 6 and Supplementary Table 5). Also, many insertions are observed which separate the PUL elements into several islands, without impairing the PUL activity, according to growth data. Examples span from the three-gene insertion specific to *B. theta* strains (BT3344-BT3347 that includes an unrelated *susCD* pair) to hundreds of genes separating *B. plebeius* orthologues into three genomic regions (Supplementary Fig. 6). Thus, orthologue

conservation is more relevant over a tight organisation when predicting growth. Searches beyond the human gut *Bacteroides* genus reveal species having all required orthologues for growth in the *Chitinophaga*, *Cytophaga* and Sphingobacteria classes (some from human skin, chicken gut and soil environments). Notably, a tightly packed PUL was found in *Propebela bergensis* that even incorporates BT4410[PL33] within the core PUL (Supplementary Fig. 6).

## Discussion

PUL$_{CS/DS/HA}$ performs a highly co-ordinated degradation of GAGs but in a contrasting way to the previously described PUL$_{Hep}$, which degraded H/HS. PUL$_{Hep}$ encodes four PLs all of which displayed significant processivity with variable sulfation tolerances[24]. PUL$_{CS/DS/HA}$, however, utilises endo–exo synergy in the periplasm between BT3350[PL8] and BT3324[PL8] for the degradation of CS-A, CS-C and DS. The PUL also deploys an endo-acting O4 sulfatase, BT3349-S1_27[4S-sulf], the rationale behind its endo mode of action, however, is not obvious. BT3349-S1_27[4S-sulf] may enable BT4410[PL33] to contribute to the degradation of CS-A and DS, where O4 sulfation would be inhibitory. A surprising finding was the important role BT4410[PL33] plays in DS metabolism within the *Bacteroides* genus, a result that could not be predicted. The anchoring of BT3349-S1_27[4S-sulf] to the periplasmic side of the membrane may localise it close to the SusC and where it could remove O4 sulfation as glycans enter the periplasm. Alternatively, it could be localised close to BT3334[HTCS] and BT3348[GH88], neither of which can utilise O4 sulfated disaccharides[13] and are also membrane associated. This may allow O4 sulfated disaccharides to be desulfated close to the HTCS before BT3348[GH88] can degrade them. BT1596-S1_9[2S-sulf] must remove any 2S sulfation present for BT3348[GH88] to act. BT3333-S1_15[6S-sulf] then removes the O6 sulfation of GalNAc leaving the final degradation products as GalNac and 4-deoxy-L-threo-5-hexosulose-uronate. For HA metabolism BT4410[PL33] is the only lyase required due to its endo-processive mode of action, and after BT3348[GH88] has acted the final products would be GlcNAc and 4-deoxy-L-threo-5-hexosulose-uronate. 4-deoxy-L-threo-5-hexosulose-uronate is able to feed directly into metabolism in the cytoplasm[25] whilst both GlcNAc and GalNAc require phosphorylation before further metabolism. In contrast to PUL$_{Hep}$, which encodes the GlcNAc kinase BT4654[ROK], PUL$_{CS/DS/HA}$ does not encode for any kinases. Therefore, phosphorylation of GalNAc and GlcNAc, must be completed by constitutively active kinases. Support for a constitutively active GlcNAc kinase is that deletion of BT4654[ROK] in PUL$_{Hep}$ has no growth defect on H/HS[24]. In addition, no differentially expressed kinases are observed during CS growth.

Although the extracellular glycan apparatus appears to confer no advantage in mono-cultured systems, it may be important in a multi-organism, competitive, environment. When *B. theta*, *B. caccae* and *B. cellulosilyticus* were co-cultured on CS for 5 days it was observed that *B. cellulosilyticus* CFUs were ≤100-fold lower than *B. theta* and *B. caccae* and ≤100 than *B. cellulosilyticus* when mono-cultured[26]. *B. cellulosilyticus* lacks any surface glycan apparatus whilst it is present in both *B. theta* and *B. caccae*.

Goodman et al.[27] identified, through transposon mutagenesis, genes in *B. theta* which were important for colonisation of the mouse gut. Strains with insertion in *bt3348* showed a marked decrease in output compared with input. In addition, when mice were colonised with an artificial microbiome of multiple Bacteroidetes species (including transposon mutagenised *B. theta* strains), *B. theta* strains with insertions in *bt3330*

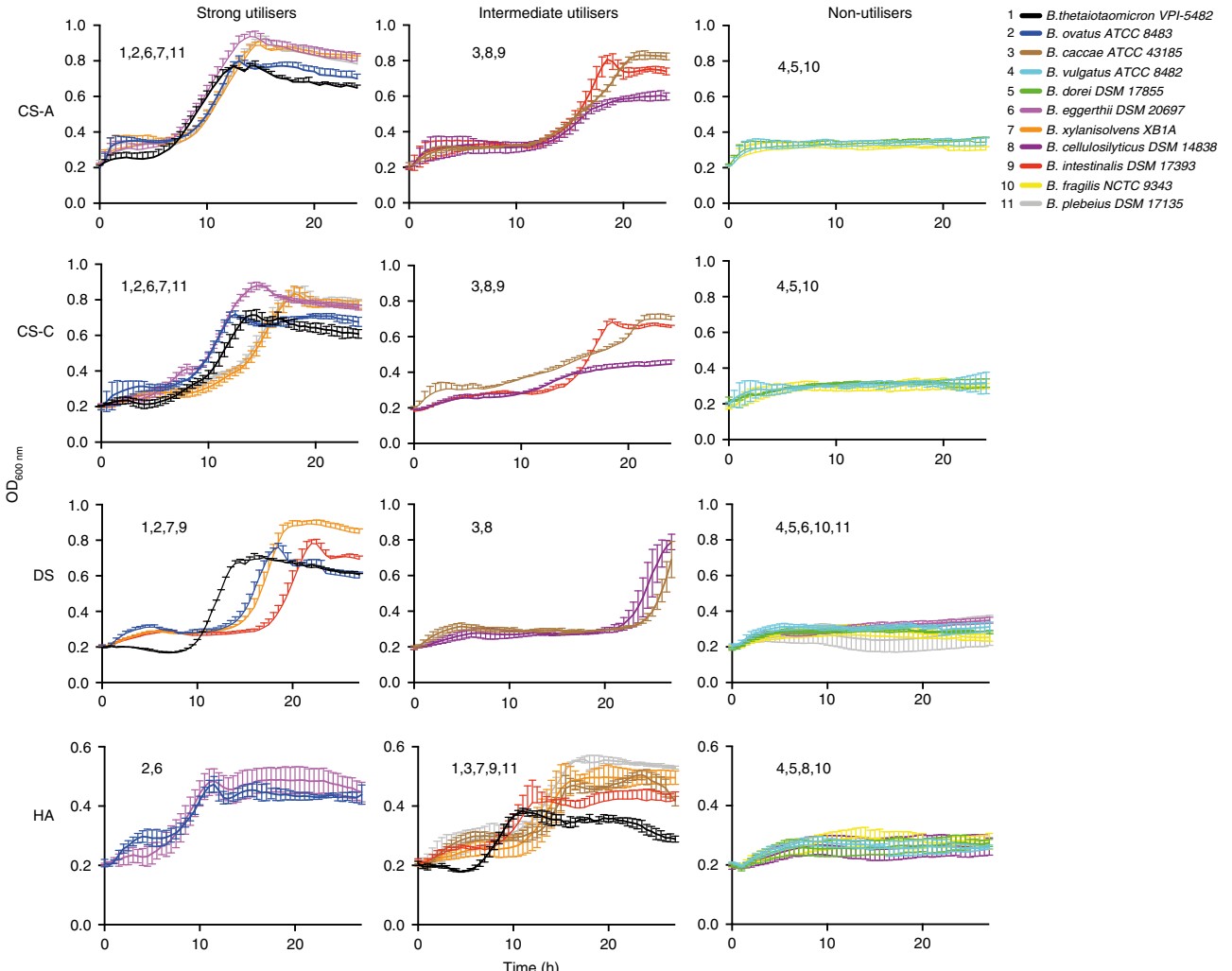

**Fig. 6 Growth curves of 11 bacteroides species on CS-A, DS, CS-C and HA.** The different species were divided into strong, intermediate and non-utilisers based on analysis of growth curve rates and final optical densities. Data represent technical triplicates and source data are provided in the source data file labelled as strain growths.

and *BT3349-S1_27* were much reduced in bacterial number compared with monocolonized mice. Thus, CS, DS and HA metabolism are under increased selection pressure in a competitive environment with other Bacteroidetes species. Conversely, *bt3328* mutants showed increased bacterial numbers, compared with monocolonized mice, when colonised with other Bacteroidetes, Firmicutes and Actinobacteria.

Indomethacin release, in the gut, from cross-linked CS is dependent upon biodegradation by the caecal contents[28]. The detailed biochemical analysis of the extracellular glycan apparatus in this study may facilitate strategies to more tightly control drug release to the digestive tract using cross-linked CS. The structural data of BT3349-S1_27[4S-sulf] shows how endo-sulfatase activity has evolved and could be used to predict/guide engineering of sulfatase function. The active site of BT3333-S1_15[6S-sulf] reveals the specificity determinants for Gal over Glc-configured substrates when compared with the GlcNAc6S sulfatase BT4656-S1-11[6S-sulf]. These data highlight how much more highly variable the active sites of related sulfatase are than could be predicted from the nature of their respective substrates. The knowledge of the highly variable nature of the glycone-binding site of sulfatases may help design future glycosulfate-specific inhibitors. This is of timely relevance because of the recently revealed role of sulfatase action in eliciting colitis in a colitis sensitive mouse model[17].

## Conclusions

The data presented here reveal how the HGM has evolved a core GAG utilisation machinery (PUL$_{CS/DS/HA}$) to target diverse GAGs in the human gut. PUL$_{CS/DS/HA}$ components included orphan, physically connected and constitutively active genes with complementary and in some cases unique activities which together enable the complete metabolism of diverse GAGs. PUL$_{CS/DS/HA}$ components are widely conserved in *Bacteroides* and thus presents a general model of CS, DS and HA metabolism, which can be applied to other members of the HGM.

## Methods

*Sources of carbohydrates*. Carbohydrates were from Sigma, carbosynth or Dextra Laboratories. All other chemical reagents were purchased from Sigma.

*Bacteroides culture and genetic manipulation*: Various Bacteroides strains and *B. theta* VPI-5482 were cultured anaerobically in a Whitley A35 Workstation (Don Whitley) at 37 °C in either tryptone yeast extract glucose medium or minimal medium (MM) containing GAGs (described below). *B. theta* strains containing specific gene deletions or inactivated versions of enzymes were made by counter selectable allelic exchange as described below[18]. All genetic mutants, except Δ*bt3328*, were provided by Dr Guy E. Townsend and Prof. Eduardo Groisman[13]. The in frame mutants were generated using counter selectable allelic exchange[29]. Briefly, 1-kb fragments either side of the target gene were cloned, 'stitched together' by PCR, and ligated into the pExchange plasmid using SalI/XbaI sites. This recombinant plasmid was then conjugated into *a tdk⁻* version of *B. theta* (referred to as wildtype (WT) in this study) using the S17-1 λ

*pir Escherichia coli* strain. Initial homologous recombination, plasmid insertion, is selected for using erythromycin resistance. Then a second low-frequency recombination event, exclusion of the pExchange plasmid from the genome, is selected for using FUdR containing plates. In frame deletion mutants were detected and checked using PCR and sequencing. Insertional mutations were constructed by cloning a 750-bp sequence from the gene to be disrupted into the pKNOCK *tet* vector, followed by its introduction into the *B. theta* chromosome[30]. Growth of the WT and mutants was measured on glycans by mixing the autoclave-sterilised polysaccharides (0.5% final) with minimal media (per 100 ml was: 0.1 g of Ammonium sulfate and sodium carbonate, 0.05 g cysteine, 10 ml 1 M potassium phosphate pH 7.2, 0.1 ml of 1 mg ml$^{-1}$ vitamin K, 1 ml of 0.4 mg ml$^{-1}$ iron sulfate, 0.4 ml of 0.25 mg ml resazurin, 0.05 ml of 0.01 mg ml$^{-1}$ vitamin$_{B12}$ and 5 ml of mineral salts for defined media) and monitoring growth continuously in 96-well plates using a Biotek Epoch plate reader. Growth curves presented are averages of three technical replicates.

*Co-immunoprecipitation and western blotting*: Bacterial cells were harvested in the mid-exponential phase and bacterial pellets were resuspended in the lysis buffer (50 mm Tris, pH 7.5, 150 mm NaCl, 0.2 mm Na$_3$VO$_4$, 1% Nonidet P-40 alternative, 1 mm PMSF, 1 mm DTT, and 1× protease inhibitors (Roche Applied Science)). Lysates were sonicated followed by incubation with 1 μg of antibodies as indicated for 16 h at 4 °C, antibodies were pulled down using protein G-Sepharose beads. Samples were then run on 12.5% SDS-PAGE gels, transferred to a nitrocellulose membrane using standard buffers. Both primary and secondary antibodies were used as 1/1000 dilutions in 5% milk powder. The secondary antibody was mouse anti-rabbit IgG-HRP (sc-2357). Normal mouse IgG (sc-2025) was used as a negative control. Both antibodies were purchased from santa-cruz.

*Microscopy*: GAG grown cells were harvested at mid-exponential phase and fixed for 90 min in formalin (10% formaldehyde in PBS buffer). Primary Ab was incubated 1/500 for 2 h at room temperature (RT), followed by incubation with the secondary Ab (Santa Cruz goat anti-rabbit FITC) at 1/500 at RT for 1 h. Cells were immobilised for the microscopy by transfer on 1.2% agarose/H$_2$O pads. Fluorescence microscopy was carried out with Nikon Eclipse Ti equipped with Nikon Plan Apo 100×/1.40 NA phase contrast objective, Prime sCMOS camera (Teledyne Photometrics) and Lambda LS Xenon-arc lamp (Sutter Instruments), and Chroma 49002 filter set (Chroma Technology). Images were acquired with Metamorph 7.7 (Molecular Devices) and analysed with Fiji[31].

*Whole-cell assays*. Whole-cell assays are a direct readout of cell surface activity. After washing with PBS there is no longer any sugar transport by the TonB dependent SuSC transporter. Therefore, product profiles and activities of surface enzymes can be assessed. *B. theta* was grown in 5 mL MM with 1% (wt/vol) appropriate GAG as the sole carbon source to mid-exponential phase in glass test tubes. Cells were harvested by centrifugation at 5000 × *g* for 10 min at RT and washed in 5 mL PBS (pH 7.2) before being resuspended in 1 ml PBS. Washed cells were assayed against 10 mg mL$^{-1}$ of the appropriate substrate at 37 °C for up to 24 h. Assays were analysed by thin layer chromatography (TLC), and 2 μL each sample was spotted onto silica plates and resolved in butanol:acetic acid:water (2:1:1) running buffer. The plates were dried, and the sugars were visualised using diphenylamine stain.

*Recombinant protein production*. Genes were amplified by PCR using the appropriate primers and the amplified DNA cloned in pET28a using BamHI/XhoI or NheI/XhoI restriction sites generating constructs with either N- or C-terminal His$_6$ tags (Supplementary Table 7). Recombinant genes were expressed in *E. coli* strains BL21 (DE3) or TUNER (Novagen), containing the appropriate recombinant plasmid, and cultured to mid-exponential phase in LB supplemented with 50 μg/mL kanamycin at 37 °C and 180 rpm. Cells were then cooled to 16 °C, and recombinant gene expression was induced by the addition of 0.1 mM isopropyl β-D-1-thiogalactopyranoside; cells were cultured for another 16 h at 16 °C and 180 rpm. The cells were then centrifuged at 5000 × *g* and resuspended in 20 mM Hepes, pH 7.4, with 500 mM NaCl before being sonicated on ice. Recombinant protein was then purified by immobilised metal ion affinity chromatography using a cobalt-based matrix (Talon, Clontech) and eluted with 100 mM imidazole. For the proteins selected for structural studies (BT3333-S1_15$^{6S-sulf}$ and BT3349-S1_27$^{4S-sulf}$), another step of size-exclusion chromatography was performed using a Superdex 16/60 S200 column (GE Healthcare), with 10 mM Hepes, pH 7.5, and 150 mM NaCl as the eluent, and they were judged to be ≥95% pure by SDS-PAGE. Protein concentrations were determined by measuring absorbance at 280 nm using the molar extinction coefficient calculated by ProtParam on the ExPasy server (web.expasy.org/protparam/).

*Isothermal Calorimetry (ITC)*. The affinity of BT3329$^{SGBP}$, BT3330$^{SGBP}$ and BT3331$^{SusD}$ for oligo- and polysaccharides was quantified by ITC using a Microcal VP calorimeter. The protein sample (50 μM), stirred at 394 rpm in a 1.4-mL reaction cell, was injected with 26 × 10-μL aliquots of ligand. Titrations were carried out in 50 mM Tris-HCl buffer, pH 8.0, at 25 °C. Integrated binding heats minus dilution heat controls were fit to a single set of sites binding model to derive $K_A$, $\Delta H$ and *n* (number of binding sites on each molecule of protein) using Microcal Origin v7.0. The molar concentration of binding sites present in polysaccharides for BT3329$^{SGBP}$ was determined by altering the concentration of ligand used for regression of the isotherm until the fit yielded a value of 1 for *n*. For BT3330$^{SGBP}$ and BT3331$^{SusD}$ *n* was fixed to 1 and a ligand concentration of 2 mM was used.

*Enzyme assays*. All assays, unless stated, were carried out in 50 mM MES, pH 6.0, or Bis-Tris propane, pH 6.5, with 150 mM NaCl at 37 °C. PL activity was monitored continuously at $A_{235 nm}$ by the formation of the carbon double bond generated through the enzymes beta elimination mechanism of action. BT3334$^{GH88}$ glucuronyl hydrolase activity was determined by monitoring loss of signal at $A_{235nm}$. The kinetic parameters of BT3349-S1_27$^{4S}$ were monitored via a continuous linked assay using BT3334$^{GH88}$ and UA-GalNAc4S as the substrate. BT3334$^{GH88}$ will only hydrolyse UA-GalNAc4S after it has been converted to UA-GalNAc by BT3349-S1_27$^{4S-sulf}$ and thus loss of sulfation can be monitored indirectly by loss of signal at $A_{235nm}$ caused by BT3334$^{GH88}$ hydrolysing any UA-GalNAc produced. The ability of BT3333-S1_15$^{6S-sulf}$, BT3349-S1_27$^{4S-sulf}$ and BT1596-S1_9$^{2S-sulf}$ to remove O$_2$, O$_4$ and O$_6$ sulfation, respectively, from unsaturated CS disaccharides was also monitored qualitatively by TLC. Reaction profiles of PLs and sulfatase digestion of GAGs were monitored by HPAEC using an AD25 or a VWD, Thermofisher, absorbance detector at $A_{235nm}$ to detect the carbon–carbon double bond products, with H$_2$O, pH 3.5, as the eluent and a second eluent of H$_2$O with 3 M NaCl used to generate a linear NaCl gradient to 100% over 90 min. All enzymatic assays were performed in technical triplicate.

*Labelling of glycans*: Labelling with 2-aminobenzamide acid (2-AB) was performed using the ludgerTag 2-AB glycan labelling kit. Briefly, DMSO was added to acetic acid in a 0.75:1 ratio and 100 μl added to 5 mg of 2-AB. This mixture is then added to the reductant sodium cyanoborohydride and when fully dissolved 5 μl was added to 50 nmol of dried HA dodecasaccharide. The reaction was left at 65 °C for ~3 h. After labelling samples were cleaned up using the provided spin columns. Labelling of HA octasaccharide and decasaccharide was carried out by creating a saturated solution of 7-aminonapthalene-1,3-disulfonic acid (ANDSA) in formamide and mixing this with 500 μg of dried glycan. The mixture was left overnight at RT and the labelled glycans used the next day in a 1/5 dilution versus enzyme buffer. Saccharides labelled using the UV fluorescent tag, ANDSA were run on 33% acrylamide tris-acetate gels with tris-MES running buffer. Gels were viewed after removal from the glass plates on a UV transilluminator[32].

*Crystallisation of BT3333-S1_15$^{6S-sulf}$ and BT3349-S1_27$^{4S-sulf}$*. After purification, BT3333-S1_15$^{6S-sulf}$ and BT3349-S1_27$^{4S-sulf}$ were carried forward in the same eluent as used for the size-exclusion chromatography. All proteins were then concentrated in centrifugal concentrators with a molecular mass cutoff of 30 kDa. Sparse matrix screens were set up in 96-well sitting drop TTP Labtech plates (400-nL drops). Initial BT3333-S1_15$^{6S-sulf}$ apo crystals were obtained at 20 mg/mL in 20% PEG 3350 and KCl. For BT3349-S1_27$^{4S-sulf}$ ligand-bound initial hits were obtained at 20 mg mL$^{-1}$ in 20% PEG 3350 and 0.2 M K acetate. BT3349-S1_27$^{4S-sulf}$ crystals were soaked overnight with 10 mM of a CS tetrasaccharide prior to fishing. BT3333-S1_15$^{6S-sulf}$ ligand-bound forms were crystallised at 20 mg mL$^{-1}$ with 10 mM GlaNAc6S in 30% PEG 4000, 200 mM MgCl$_2$, and 0.1 M Tris, pH 8.5. All crystals were cryo-cooled with the addition of 20% PEG 400. Data were collected at Diamond Light Source (Oxford) on beamlines I24 and I04-1 (0.98 Å) at 100 K. The data were integrated with XDS[33] and scaled with Aimless[34,35]. Five percent of observations were randomly selected for the R$_{free}$ set. The phase problem was solved by molecular replacement using the program automated molecular replacement server Balbes[36] for BT3349-S1_27$^{4S-sulf}$. The phase problem for BT3333-S1_15$^{6S-sulf}$ was initially solved using Molrep[37] and BT1624 (unpublished structure) as the search model. This gave a partial solution, which could not be fully solved due to tNCS and/or twinning. A good enough model of BT3333-S1_15$^{6S-sulf}$ was constructed to be used to solve the phase problem for subsequent collected data, which showed no tNCS and/or twinning defects. Models underwent recursive cycles of model building in Coot[38] and refinement cycles in Refmac5[39]. The models were validated using Coot[38] and MolProbity[40]. Structural Figures were made using Pymol (the PyMOL Molecular graphics system, Version 2.0 Schrodinger, LLC) and all other programmes used were from the CCP4 suite[41]. The data processing and refinement statistics are reported in Supplementary Table 6.

*Site-directed mutagenesis*. Site-directed mutagenesis was conducted using the PCR-based QuikChange kit (Stratagene) according to the manufacturer's instructions using the appropriate plasmid as the template and appropriate primer pairs.

*Comparative genomics analysis*. Similar PULs were searched in >900 Bacteroidetes genomes as follows. First, we identified orthologous proteins to PUL$_{CS/DS/HA}$ based on reciprocal best BlastP results between *B. theta* and each species. Orthologous proteins were mapped to their PUL predictions in PULDB to identify loci gathering several orthologues. For the 11 species tested for growth, PUL predictions were manually refined (enlarged) for boundaries to cover the whole PUL$_{CS/DS/HA}$ region which included variable insertions between the core operon and BT3324 or BT3349-S1_27-BT3350 orthologues.

**Reporting summary**. Further information on research design is available in the Nature Research Reporting Summary linked to this article.

## Data availability
The datasets generated during and/or analysed during the current study are available from the corresponding author on reasonable request. Protein structure data concerned with the structures solved in this work, 6S21 and 6S20, have been deposited with the protein data bank (PDB). Data underlying Figs. 3, 6, Supplementary Figs. 2–4 and Supplementary Tables 2 and 4 have been provided in a source data folder.

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

## Acknowledgements

The authors would like to thank Diamond Light Source for beamtime and the staff of beamlines I04 and I24 for assistance with crystal testing and collection. We would also like to thank Prof. Eduardo Groisman and Dr Guy Townsend for providing *B. theta* strains. We would also like to thank Carl Morland for his expert technical assistance. This work was supported by start up funds provided by Liverpool University to A.C. The work was also supported by the MRC CiC award MC_PC_17166 held by E.A.Y and a Wellcome ISSF grant 204822/Z/16/Z awarded to U.L.M. A.B. was supported on an ERC advanced award (Grant 322820).

## Author contributions

D.N. and A.C. performed enzymology and binding kinetics. D.N. and A.C. performed protein localisation. D.N. performed bacterial growths. H.S. carried out microscopy. N.T and B.H. performed bioinformatic analysis. A.C and A.B. carried out X-ray crystallography experiments. A.C and U.L.M. carried out Co-IP experiments. A.C. and E.A.Y. carried out sugar-labelling experiments. A.C. conceptualised the work and wrote first draft. A.C and N.T prepared Figures. All authors contributed to revision and editing of the paper.

## Competing interests

The authors declare no competing interests.
