## [Peer Review File · Nature Communications]

Reviewers' comments:

Reviewer #1 (Remarks to the Author):

The manuscript by Didier Ndeh et al. describes an integrated study of all gene products belonging to a specific gene location (PUL) that targets diverse GAGs, originating from *B. thetaiotaomicron* (Bt), a prominent gut species. The experimental work includes biochemical, structural as well as functional analyses of gene products upregulated in this PUL and that, in conjunction with two distal genes, allow Bt to grow on various substrates belonging to the GAG family. Altogether, the paper is very well written and the experiments, covering a large range of techniques, give conclusive and original results. Key results are the identification and characterization of surface glycan binding proteins (SGBP) that have not been described before. In addition the specificity and reaction conditions for several PL8, PL29 and PL33 polysaccharide lyases and two sulfatases are biochemically depicted. Overall, the study explores the substrate specificities of all of the enzymes/proteins, compares growth of various knock-out mutants of Bt and other gut Bacteroides species to understand in detail the cascade of reactions leading to the degradation and uptake of GAGs. The article is thus of high interest, will certainly bring valuable information to readers largely beyond the microbial glycoscience community, and I therefore recommend publication after having taken into account the comments and recommendations listed below.

Comments

1. Abstract Page 2, line 42 change “...of the how the ‘sulfation problem’ is handled” to “...of how the ‘sulfation problem’ is handled”

2. Results Page 3, lines 86-95. A link to Figure 1B is missing for a (non specialist) reader to fully be able to assimilate the chemical structure description of these GAGs.

3. Results page 4, lines 109 and following : the abbreviation “SGBP” should be written out (surface glycan binding protein) in this section at least once for explanation.

4. Results page 4, paragraph on periplasmic glycan degradation apparatus: the data presented for these proteins are scarce; only interactions with ligands were measured by ITC, but what about protein-protein interactions? Do they interact with each other? Or with SusD or other outer membrane located proteins/enzymes of the PUL? When looking at the model of the GAG degradation PUL in Figure 1, the SGBP are sketched as two interacting black circles but separated

from the rest....where does this hypothesis come from? Also, why weren't these proteins included in the knockout mutant studies to assess some relevant functional data?

5. Results page 4, last paragraph lines 137 to 141 "The N-terminal regions of BT3328PL29 and BT3329SGBP (~300 amino-acids) share remote sequence similarity. Furthermore, this is shared with SGBP positioned genes from other PULs. This could correspond to a general 'spacer' domain, allowing extracellular proteins to extend out of the capsular envelope."

This statement is not very clear. Does the sequence similarity cover the entire 300 N-terminal amino-acids? Is the 'spacer domain' 300 aa large? Or is a certain stretch within this domain conserved? What about similarities or differences of SGBP BT3329 and BT3330 with "SGBP positioned genes from other PULs" – does this only concern the 300 aa? Are there any other similarities? This should be elaborated or explained a bit more in detail.

5. Results page 4, lines 125-126 : The reason for the divergence in activity observed for BT3328 described here and in reference 20 by the same authors should be at least commented in the supplemental material.

6. Results page 4, lines 135-137 and Figure S2C : "When BT3328PL29 was incubated with a HA dodecasaccharide, labelled at its reducing end with a fluorescent tag, a labelled trisaccharide was produced indicating that BT3328PL29 cleaves glycans from the reducing end (Fig S2C)."

To fully understand and follow the authors' reasoning how they conclude that BT3328-PL29 cleaves from the reducing end, the arguments should be a bit more detailed (in the legend of figure S2C, for example by explaining the difference of pattern produced by PL29 on un-labelled and labelled DP12 lanes 7 and 8?)

7. Results page 6, lines 209-210 : The subfamily numbers of sulfatases following the "Sulfatlas" nomenclature should be given at least once (or included in the naming). Since "Sulfatlas" has an option to blast against the database this is straight forward (website location <http://abims.sb-roscoff.fr/sulfatlas/>).

Figure 1 A the distal BT3350 is erroneously labelled PL12 instead of PL8 (above the arrow).

Figure legend S1 : please correct the spelling of "relevant glycosaminglcan" to glycosaminoglycan ;Also the explanation of NQ and NB should be moved from FigS2 into the legend of FigS1.

Some typos:

Line 66-67 “to be a high priority carbon sources” should be either “to be high priority carbon sources” or “to be a high priority carbon source”.

Line 338 “localise it close the SusC ..” should read “localise it close to the SusC..”

Line 399 in “DS and HA metabolism which, can be”the comma is misplaced

Reviewer #2 (Remarks to the Author):

Review of The metabolism of multiple glycosaminoglycans by the human gut microbiota is orchestrated by a versatile core genetic locus

This study dissects the polysaccharide utilization loci for the usage of glycosaminoglycans in *Bacteroides thetaiotaomicron*. Overall this is well executed study that provides important information as to how *Bacteroides* and other members of the human gut microbiome utilizes host derived complex carbohydrates. It also provides a blueprint for determining GAG consumption by other *bacteroides* species or other microbes, which will be highly useful as additional sequences of bacteria become available. My only major criticism is that it would be useful if the authors walked through the model depicted in Fig 1b so that readers that are less familiar with how PULs operate and specifically what was known about this PUL prior to this study.

Minor comments:

BT3348 is referred to as BT3488 throughout the manuscript.

BT3350 is listed as a PL8 in the manuscript but as a PL12 in Figure 1a

Line 149 refers to Fig 3 instead of Fig 2

B. eggerthii is misspelled throughout the manuscript

Prevotella bergensis is also misspelled

N Comms Reviewers' comments:

Reviewer #1 (Remarks to the Author):

The manuscript by Didier Ndeh et al. describes an integrated study of all gene products belonging to a specific gene location (PUL) that targets diverse GAGs, originating from *B. thetaiotaomicron* (Bt), a prominent gut species. The experimental work includes biochemical, structural as well as functional analyses of gene products upregulated in this PUL and that, in conjunction with two distal genes, allow Bt to grow on various substrates belonging to the GAG family. Altogether, the paper is very well written and the experiments, covering a large range of techniques, give conclusive and original results. Key results are the identification and characterization of surface glycan binding proteins (SGBP) that have not been described before. In addition the specificity and reaction conditions for several PL8, PL29 and PL33 polysaccharide lyases and two sulfatases are biochemically depicted. Overall, the study explores the substrate specificities of all of the enzymes/proteins, compares growth of various knock-out mutants of Bt and other gut *Bacteroides* species to understand in detail the cascade of reactions leading to the degradation and uptake of GAGs. The article is thus of high interest, will certainly bring valuable information to readers largely beyond the microbial glycoscience community, and I therefore recommend publication after having taken into account the comments and recommendations listed below.

Comments

1. Abstract Page 2, line 42 change "...of the how the 'sulfation problem' is handled" to "...of how the 'sulfation problem' is handled"

This has been corrected. Page 1 line 33.

2. Results Page 3, lines 86-95. A link to Figure 1B is missing for a (non specialist) reader to fully be able to assimilate the chemical structure description of these GAGs.

This has been corrected. Page 3, line 89.

3. Results page 4, lines 109 and following : the abbreviation "SGBP" should be written out (surface glycan binding protein) in this section at least once for explanation.

This has been corrected. Page 3 line 101 and page 4 line 114.

4. Results page 4, paragraph on periplasmic glycan degradation apparatus: the data presented for these proteins are scarce; only interactions with ligands were measured by ITC, but what about protein-protein interactions? Do they interact with each other? Or with SusD or other outer membrane located proteins/enzymes of the PUL? When looking at the model of the GAG

degradation PUL in Figure 1, the SGBP are sketched as two interacting black circles but separated from the rest....where does this hypothesis come from? Also, why weren't these proteins included in the knockout mutant studies to assess some relevant functional data?

page 5 line 151 .Protein-protein interactions have now been assessed both in an in vitro and in vivo context and the data now constitutes a new Fig S3 and are described. Figure 1 has been modified accordingly to reflect the new data. The SGBP KOs were not made as the focus of the paper was on the enzymatic process. This work is, however, on going.

5. Results page 4, last paragraph lines 137 to 141 "The N-terminal regions of BT3328PL29 and BT3329SGBP (~300 amino-acids) share remote sequence similarity. Furthermore, this is shared with SGBP positioned genes from other PULs. This could correspond to a general 'spacer' domain, allowing extracellular proteins to extend out of the capsular envelope."

This statement is not very clear. Does the sequence similarity cover the entire 300 N-terminal amino-acids? Is the 'spacer domain' 300 aa large? Or is a certain stretch within this domain conserved? What about similarities or differences of SGBP BT3329 and BT3330 with "SGBP positioned genes from other PULs" – does this only concern the 300 aa? Are there any other similarities? This should be elaborated or explained a bit more in detail.

A new paragraph has been added to clarify this point. Page 4 line 140.

5. Results page 4, lines 125-126 : The reason for the divergence in activity observed for BT3328 described here and in reference 20 by the same authors should be at least commented in the supplemental material.

The data presented for BT3328 was assessed kinetically by quantitative UV spectroscopy and qualitatively by HPAEC and TLC. Where activity was low spectroscopically it was low when observed by HPAEC and TLC. This provides three methods which all agree on the activity of BT3328 and we are therefore confident of the data. No authors on this manuscript generated the kinetic data in reference 20, therefore we are not well positioned to accurately explain the divergence in activities.

6. Results page 4, lines 135-137 and Figure S2C : "When BT3328PL29 was incubated with a HA dodecasaccharide, labelled at its reducing end with a fluorescent tag, a labelled trisaccharide was produced indicating that BT3328PL29 cleaves glycans from the reducing end (Fig S2C)."

To fully understand and follow the authors' reasoning how they conclude that BT3328-PL29 cleaves from the reducing end, the arguments should be a bit more detailed (in the legend of figure S2C, for example by explaining the difference of pattern produced by PL29 on un-labelled and labelled DP12 lanes 7 and 8?)

A more thorough explanation of the data has been added in the supplemental page 1, line 5. In addition new data utilising a labelled octasaccharide and decasaccharide has been produced which allows simpler analysis of the enzymes

mode of action.

7. Results page 6, lines 209-210 : The subfamily numbers of sulfatases following the "Sulfatlas" nomenclature should be given at least once (or included in the naming). Since "Sulfatlas" has an option to blast against the database this is straight forward (website location <http://abims.sb-roscoff.fr/sulfatlas/>).

The subfamily number has been added in all instances the sulfatases are mentioned. A modified paragraph has also been inserted at page 7 line 232.

Figure 1 A the distal BT3350 is erroneously labelled PL12 instead of PL8 (above the arrow).

This has been corrected in Figure 1A.

Figure legend S1 : please correct the spelling of "relevant glycosaminglcan" to glycosaminoglycan ;

This has been corrected. Page 4 Supplemental information.

Also the explanation of NQ and NB should be moved from FigS2 into the legend of FigS1.

Definitions for NQ and NB have been added to the legend of Table S1.

Some typos:

Line 66-67 "to be a high priority carbon sources" should be either "to be high priority carbon sources" or "to be a high priority carbon source".

This has been corrected. Page 2 line 58

Line 338 "localise it close the SusC .." should read "localise it close to the SusC.."

This has been corrected. Page 11 line 363.

Line 399 in "DS and HA metabolism which, can be"the comma is misplaced

This has been corrected. Page 13 line 423.

Reviewer #2 (Remarks to the Author):

Review of The metabolism of multiple glycosaminoglycans by the human gut microbiota is orchestrated by a versatile core genetic locus

This study dissects the polysaccharide utilization loci for the usage of glycosaminoglycans in *Bacteroides thetaiotaomicron*. Overall this is well executed study that provides important information as to how *Bacteroides* and other members of the human gut microbiome utilizes host derived complex

carbohydrates. It also provides a blueprint for determining GAG consumption by other bacteroides species or other microbes, which will be highly useful as additional sequences of bacteria become available. My only major criticism is that it would be useful if the authors walked through the model depicted in Fig 1b so that readers that are less familiar with how PULs operate and specifically what was known about this PUL prior to this study.

A short paragraph has been added describing the paradigm of how PULs operate. Page 3 line 99.

Minor comments:

BT3348 is referred to as BT3488 throughout the manuscript.

This has been corrected and highlighted in the manuscript.

BT3350 is listed as a PL8 in the manuscript but as a PL12 in Figure 1a

This has been corrected in figure 1a

Line 149 refers to Fig 3 instead of Fig 2

Line 149 is referring to HPAEC traces found in Fig. 2 not the growth curves in Fig 3.

B. eggerthii is misspelled throughout the manuscript

This has been corrected and highlighted in the text

Prevotella bergensis is also misspelled

This has been corrected and highlighted in the text.

REVIEWERS' COMMENTS:

Reviewer #1 (Remarks to the Author):

The authors have satisfyingly answered all my questions and done the changes requested, I therefore support publication of the manuscript in its actual form. I have no additional comments, this is a very nice and thoroughly prepared piece of work!

Reviewer #2 (Remarks to the Author):

Thank you for addressing my comments.

I think there is some confusion of the referencing of Figure 3 on what is now line 172. The current version of the manuscript references Fig. 3 (the growth curves) when it should reference Fig 2, the HPAEC traces (as you yourself state in the rebuttal).

REVIEWERS' COMMENTS:

Reviewer #1 (Remarks to the Author):

The authors have satisfyingly answered all my questions and done the changes requested, I therefore support publication of the manuscript in its actual form. I have no additional comments, this is a very nice and thoroughly prepared piece of work!

We thank the reviewer for the thorough review and were glad we could answer all of their questions.

Reviewer #2 (Remarks to the Author):

Thank you for addressing my comments.

I think there is some confusion of the referencing of Figure 3 on what is now line 172. The current version of the manuscript references Fig. 3 (the growth curves) when it should reference Fig 2, the HPAEC traces (as you yourself state in the rebuttal).

The reviewer is indeed correct that we misunderstood. We have now corrected this error and changed the mentioning of figure 3 on line 172 (now line 192 in) to figure 2. We are happy we could address the reviewers comments and thank the reviewer for correcting our mistake a second time.